# Placental Complement Activation in Fetal and Neonatal Alloimmune Thrombocytopenia: An Observational Study

**DOI:** 10.3390/ijms22136763

**Published:** 2021-06-23

**Authors:** Thijs W. de Vos, Dian Winkelhorst, Hans J. Baelde, Kyra L. Dijkstra, Rianne D. M. van Bergen, Lotte E. van der Meeren, Peter G. J. Nikkels, Leendert Porcelijn, C. Ellen van der Schoot, Gestur Vidarsson, Michael Eikmans, Rick Kapur, Carin van der Keur, Leendert A. Trouw, Dick Oepkes, Enrico Lopriore, Marie-Louise P. van der Hoorn, Manon Bos, Masja de Haas

**Affiliations:** 1Division of Neonatology, Department of Pediatrics, Willem-Alexander Children’s Hospital, Leiden University Medical Center, 2300 RC Leiden, The Netherlands; e.lopriore@lumc.nl; 2Center of Clinical Transfusion Research, Sanquin Research, 1066 CX Amsterdam, The Netherlands; M.dehaas@sanquin.nl; 3Department of Experimental Immunohematology, Sanquin Research, 1066 CX Amsterdam, The Netherlands; dian.winkelhorst@gmail.com (D.W.); e.vanderschoot@sanquin.nl (C.E.v.d.S.); G.Vidarsson@sanquin.nl (G.V.); r.kapur@sanquin.nl (R.K.); 4Department of Obstetrics and Gynecology, Leiden University Medical Center, 2300 RC Leiden, The Netherlands; d.oepkes@lumc.nl (D.O.); M.L.P.van_der_Hoorn@lumc.nl (M.-L.P.v.d.H.); M.Bos@lumc.nl (M.B.); 5Department of Pathology, Leiden University Medical Center, 2300 RC Leiden, The Netherlands; J.J.Baelde@lumc.nl (H.J.B.); K.L.Dijkstra@lumc.nl (K.L.D.); riannevanb@hotmail.com (R.D.M.v.B.); L.van_der_Meeren@lumc.nl (L.E.v.d.M.); 6Department of Pathology, University Medical Center Utrecht, 3508 GA Utrecht, The Netherlands; p.g.j.nikkels@umcutrecht.nl; 7Department Immunohematology Diagnostics, Sanquin Diagnostic Services, 1066 CX Amsterdam, The Netherlands; L.Porcelijn@sanquin.nl; 8Department of Immunology, Leiden University Medical Center, 2300 RC Leiden, The Netherlands; m.eikmans@lumc.nl (M.E.); C.van_der_Keur@lumc.nl (C.v.d.K.); L.A.Trouw@lumc.nl (L.A.T.); 9Department of Hematology, Leiden University Medical Center, 2300 RC Leiden, The Netherlands

**Keywords:** fetal neonatal alloimmune thrombocytopenia, alloimmunization during pregnancy, placental dysfunction, fetal growth restriction, histopathology placenta, classical pathway complement activation

## Abstract

Fetal and neonatal alloimmune thrombocytopenia (FNAIT) is a disease that causes thrombocytopenia and a risk of bleeding in the (unborn) child that result from maternal alloantibodies directed against fetal, paternally inherited, human platelet antigens (HPA). It is hypothesized that these alloantibodies can also bind to the placenta, causing placental damage. This study aims to explore signs of antibody-mediated placental damage in FNAIT. We performed a retrospective study that included pregnant women, their newborns, and placentas. It comprised 23 FNAIT cases, of which nine were newly diagnosed (14 samples) and 14 were antenatally treated with intravenous immune globulins (IVIg) (21 samples), and 20 controls, of which 10 had anti-HLA-class I antibodies. Clinical information was collected from medical records. Placental samples were stained for complement activation markers (C1q, C4d, SC5b-9, and mannose-binding lectin) using immunohistochemistry. Histopathology was examined according to the Amsterdam criteria. A higher degree of C4d deposition was present in the newly diagnosed FNAIT cases (10/14 samples), as compared to the IVIg-treated FNAIT cases (2/21 samples, *p* = 0.002) and anti-HLA-negative controls (3/20 samples, *p* = 0.006). A histopathological examination showed delayed maturation in four (44%) placentas in the newly diagnosed FNAIT cases, five (36%) in the IVIg-treated FNAIT cases, and one in the controls (NS). C4d deposition at the syncytiotrophoblast was present in combination with low-grade villitis of unknown etiology in three newly diagnosed FNAIT cases that were born SGA. We conclude that a higher degree of classical pathway-induced complement activation is present in placentas from pregnancies with untreated FNAIT. This may affect placental function and fetal growth.

## 1. Introduction

Fetal and neonatal alloimmune thrombocytopenia (FNAIT) is the leading cause of thrombocytopenia and bleeding tendency in otherwise healthy and term-born infants. FNAIT is caused by maternal alloantibodies directed against paternally inherited human platelet antigens (HPA) [1]. Immunoglobulin G (IgG) class alloantibodies cross the placenta and bind to fetal platelets, resulting in thrombocytopenia and risk of bleeding. Bleeding complications in FNAIT are often caused by HPA-1a specific alloantibodies and vary from minor skin bleeding to severe intracranial hemorrhage (ICH), leading to lifelong neurological impairment or death [2]. Administration of intravenous immune globulins (IVIg) to the mother during pregnancy can prevent the hemorrhagic complications due to FNAIT [3]. However, since FNAIT is predominantly diagnosed after birth, prenatal treatment can usually only be administered in subsequent pregnancies.

The HPA-1a/HPA-1b alloantigenic epitopes are formed due to a single nucleotide substitution (C29523T), resulting in a Leu33Pro amino acid polymorphism within the integrin β3 subunit. Platelets express high levels of HPA-1a at the integrin β3 on the fibrinogen receptor (α2β3, glycoprotein IIbIIIa, CD41/CD61) and at a much lower level on the vitronectin receptor (αvβ3, CD51/CD61). Platelet-directed antibodies may lead to antibody-mediated destruction of sensitized cells, e.g., leading to thrombocytopenia, and also impair the function of these integrins [4,5,6]. Trophoblast cells and endothelial cells express high levels of the vitronectin receptor, and interference of HPA-1a antibodies with their cellular function is likely involved in the bleeding tendency [7]. In a Norwegian cohort study [8], the presence of anti-HPA-1a in maternal serum was associated with a reduced birthweight in infants diagnosed with FNAIT, which was also reported in other cohorts [9,10]. Furthermore, immune-induced placental dysfunction was observed in murine FNAIT models, in which the mice had high levels of anti-β3 antibodies and showed fetal growth restriction (FGR) and miscarriages [11]. Yougbaré et al. [12] showed abnormal placental vascularization and poor placental perfusion resulting in FGR in these mice.

Antibodies in pregnancies complicated by antiphospholipid syndrome or systemic lupus erythematosus (SLE) can also bind to trophoblast cells, which results in classical pathway complement activation [13]. In these pregnancies, complement activation is associated with adverse pregnancy outcomes, such as fetal loss and children born small for gestational age [14,15]. On the basis of these observations, we hypothesized that placental classical route complement activation might also occur in FNAIT and could lead to placental dysfunction. The aim of this study was to explore the presence of classical route complement activation and histopathological abnormalities in placentas from FNAIT cases.

## 2. Results

### 2.1. Clinical Characteristics

Clinical characteristics are shown in Table 1. Newly detected FNAIT cases (Group 1) and IVIg-treated FNAIT cases (Group 2) were comparable to controls in terms of maternal age, gravidity, parity, and presence of pre-eclampsia. Gestational age at delivery was lower in both the newly detected FNAIT cases (Group 1) and IVIg-treated FNAIT cases (Group 2) as compared to controls. ICH was observed in one of the IVIg-treated FNAIT cases (Group 2); this ICH was detected by ultrasound at 27 weeks’ gestation, while the IVIg treatment was planned to start at 28 weeks. Placental weight was below the 10th percentile in four (44%) newly diagnosed FNAIT cases, one (7%) FNAIT IVIg-treated case, and none of the controls (*p* = 0.022). Three of these four cases were SGA.

### 2.2. C4d Deposition on Syncytiotrophoblast Is Increased in Newly Detected FNAIT Cases

Placental complement deposition was scored semi-quantitatively; representative examples are shown in Appendix A. More C4d deposition was present at the syncytiotrophoblast in newly diagnosed FNAIT cases (Group 1; 10/14 samples), as compared to IVIg-treated FNAIT cases (Group 2; 2/21 samples; *p* = 0.002) and controls without anti-HLA class I antibodies (Group 4; 3/20 samples; *p* = 0.006; Figure 1A). C4d can be regarded as the footprint of complement activation, which may be either induced through the classical route (e.g., antibody-mediated) or lectin pathway activation. C1q deposition, a marker for the classical pathway of complement activation, was equally present in all study groups in either a focal (20–45%) or diffuse (20–57%) staining pattern (Figure 1B). In 11 (92%) of the samples from FNAIT cases (Group 1 and 2) with C4d deposition, C1q deposition was also present. Deposition of MBL, which indicates complement system activation via the lectin route, was observed only once (Figure 1C). The staining pattern for ongoing complement activation resulting in the formation of a membrane attack complex (MAC) did not differ between the study groups (Figure 1D). In 41% (5/12) of the samples from FNAIT cases (Group 1 and Group 2) with C4d deposition at the syncytiotrophoblast, MAC was also present. C4d deposition was present in all four placentas of newly detected FNAIT children born SGA and was only absent in the IVIg-treated FNAIT case born SGA. In the samples from FNAIT cases (Group 1 and 2), C4d deposition was equally present in placentas from term (6/6) and preterm (2/4) pregnancies (GA < 37 weeks). No relationship was observed between the mode of delivery and C4d deposition at the syncytiotrophoblast.

The presence of complement depositions in the fetal vessels is shown in Appendix A. The pattern of C4d deposition in the fetal vessels was only different between the newly diagnosed FNAIT cases (Group 1; 50% focal, 21% diffuse) as compared to the IVIg-treated FNAIT cases (Group 2; 62% focal, 10% diffuse; *p* = 0.012). C4d deposition was observed to a similar extent, though with different distribution, in controls. C1q deposition in the fetal vessels was more often diffuse (25% focal, 75% diffuse) in the control group with anti-HLA class I antibodies (Group 3) as compared to the newly detected FNAIT cases (Group 1; 71% focal, 21% diffuse; *p* = 0.003). We did not observe other significant differences.

### 2.3. Placenta Maturation Was Delayed in Newly Diagnosed FNAIT Cases

Histopathological findings in the placental tissue are described in Table 2. In four placentas of newly diagnosed FNAIT cases (Group 1; 4/9 cases; 44%) and five of the IVIg-treated FNAIT cases (Group 2; 5/14 cases; 36%), a delayed placental maturation was observed, whereas this was not observed in any of the controls (*p* = 0.118). In three of the nine cases (Group 1 and 2; 3/9 cases; 33%) with delayed maturation, the presence of C4d at the syncytiotrophoblast was detected. In one placenta of the newly diagnosed FNAIT cases with delayed maturation, the infant was born SGA. In both Group 1 and Group 2, three cases showed maternal vascular malperfusion (*p* = 0.156). Low-grade villitis of unknown etiology was observed in all groups. In all three cases with low-grade villitis of unknown etiology in the group of newly diagnosed FNAIT cases (Group 1), C4d deposition at the syncytiotrophoblast was observed and these three infants were all born SGA. Mild signs of fetal hypoxia were seen in two of the newly diagnosed FNAIT cases (Group 1; 2/9 cases; 22%) and in one of the IVIg-treated FNAIT cases (Group 2, 1/14 cases; 7%; *p* = 0.475). Decidual arteriopathy, chronic intervillositis, and vascular necrosis were not seen in any of the cases nor in the controls. An overview of clinical characteristics, complement deposition, and histopathological findings for all individual FNAIT cases is listed in Appendix A.

## 3. Discussion

### 3.1. Main Findings

The aim of this study was to explore complement activation and histopathological anomalies in the placenta of FNAIT cases. We observed an increase in C4d deposition at the syncytiotrophoblast in newly diagnosed FNAIT cases (10/14 samples), as compared to antenatally IVIg-treated FNAIT cases (2/21 samples) and healthy controls (*p* = 0.006). C4d deposition was present at the syncytiotrophoblast in all infants born SGA with newly diagnosed FNAIT, but not in the SGA newborn from a pregnancy treated with IVIg. Histopathological examination revealed a delayed maturation of the placenta parenchyma in 44% of the newly diagnosed FNAIT cases, 36% of the IVIg-treated FNAIT cases, and only one of the controls (10%) (NS). Both low-grade villitis of unknown etiology and C4d deposition at the syncytiotrophoblast were observed in three out of four newly diagnosed FNAIT cases that were born SGA.

### 3.2. Strengths and Limitations

As a result of the retrospective nature of our study, we unfortunately had limited availability of placental tissue from newly diagnosed FNAIT cases. FNAIT is often diagnosed after delivery. The need for a clinical indication to send the placenta to pathology may have led to selection bias. This was not the case for the IVIg-treated FNAIT cases and our control cohort, as these placentas were routinely stored for examination. Differences in the inclusion of our groups made it challenging to decipher whether alterations in complement deposits were due to less severe FNAIT or due to the effect of IVIg treatment. The inclusion of newly diagnosed FNAIT cases was based on the nationwide Dutch retrospective cohort study (2006–2017) [9]. A strength of our study is that the pathology departments of all referring hospitals were contacted to retrieve the maximum amount of placenta material available. We previously reported that 22% of the infants in the nationwide Dutch retrospective cohort were SGA [9]; therefore, the number of SGA in this newly diagnosed FNAIT cohort (44%) does not appear to be an overrepresentation. Furthermore, Buurma et al. [16] found a comparable amount of C4d depositions in IUGR and control placentas. Thus, SGA was suggested to not be associated with C4d deposition in the placenta. A limitation of our study is that the baseline characteristics between groups were slightly different. FNAIT cases had a lower gestational age at birth, likely due to the routine near-term induction in this group. When comparing term and preterm cases, however, our main outcome measure of C4d deposition was found to be equally present in both groups. Similar to our study, Buurma et al. [16] did not report an association between placental complement activation and the mode of delivery.

### 3.3. Interpretation

In line with other antibody-mediated disorders, such as antiphospholipid syndrome and SLE, we hypothesized that anti-HPA-1a binding could lead to classical pathway complement activation in the placenta during FNAIT pregnancies [13,15]. The complement system is tightly regulated in the placenta. C1q is reported to be an important factor in physiological trophoblast migration [17]. The balance between an active role of C1q and prevention of ongoing complement activation is controlled by complement regulatory proteins, which inhibit the formation of MAC on the membrane surface [18].

In the newly diagnosed FNAIT cases, a significantly higher degree of C4d deposition was observed at the maternal interface of the placenta of newly diagnosed FNAIT cases as compared to IVIg-treated FNAIT pregnancies (*p* = 0.002) and anti-HLA antibody-negative controls (*p* = 0.006). This may be related to the high IgG levels in the maternal circulation of the IVIg-treated women, which could have reduced anti-HPA-1a antibody levels [19]. We did not observe increased staining for MAC at the syncytiotrophoblast in the newly diagnosed FNAIT cases. This may be explained by the notion that the complement system is tightly regulated in the placenta by complement inhibitory proteins [18]. For instance, membrane cofactor protein (CD46), decay-accelerating protein (CD55), and MAC-inhibitory protein (CD59) inhibit the continuation of the complement cascade beyond C4b formation [20]. C4d is seen as a footprint of complement activation, whereas MAC deposition can be transient [13]. Moreover, C4d is an accepted biomarker in antibody-mediated transplant rejection and is also recognized to have a role in antibody-mediated pregnancy complications, [13] to which HPA-1a alloimmunization may be added based on our current results.

It is known that subtypes of HPA-1a-directed antibodies can bind to endothelial cells [4,6]. In contrast to the differences in complement depositions between the groups at the syncytiotrophoblast, no differences were observed in complement depositions between the fetal vessels of the FNAIT cases and the controls. Several factors may explain these differences between the syncytiotrophoblast and fetal vessels, such as the presence of lower levels of antibody titers in the fetus, lower availability of complement proteins on the fetal vessel side, or differences in the expression levels of complement regulatory proteins at the surface of endothelial cells and the syncytiotrophoblast.

Delayed placental maturation was found in 44% of the newly diagnosed cases, 36% of the IVIg-treated FNAIT cases, and only once in the control group (10%). Delayed maturation may be caused by villitis due to the binding of alloantibodies to trophoblasts, which may lead to altered growth and maturation of the terminal villi. Delayed maturation is associated with fetal hypoxia [21] and was also found in the placentas of β3-immunised mice [12]. C4d deposition and low-grade villitis of unknown etiology were present in three out of four newly diagnosed FNAIT cases born SGA. This finding suggests a possible correlation between complement deposition and villitis with fetal growth restriction in FNAIT [22]. Our sample size, however, was relatively small and low-grade villitis of unknown etiology was also present in some of the control cases, indicating that this finding should be interpreted with caution. In other cohort studies that assessed placental histopathology in FNAIT, chronic villitis, chronic chorioamnionitis, and chronic intervillositis were observed [23,24]. Aberrant histopathology was less pronounced in our FNAIT cases as compared to the findings of these studies; however, median gestational age at delivery was 34 weeks in the French study as compared to 37 weeks in our study, which may explain the histopathological differences [24].

### 3.4. Future Perspectives

In earlier studies, the expression of the β3-integrin on the syncytiotrophoblast and extravillous trophoblasts was shown [12,25]. In these studies an antibody directed against the β3-integrin was used. Eksteen et al. [26] showed binding of the HPA-1a antibody to the αvβ3 receptor on isolated trophoblasts. To further support our hypothesis that binding of anti-HPA-1a can lead to placenta damage, future research should focus on demonstrating binding of anti-HPA-1a to the syncytiotrophoblast.

On the basis of our observations of increased classical route complement activation in FNAIT placentas, in combination with the histopathological findings, it is imperative to perform additional investigations in the placentas of HPA-1a-negative women with and without alloantibodies and in infants born SGA.

## 4. Materials and Methods

### 4.1. Study Cohort and Placenta Collection

A total of 43 placentas were included in this study; they were categorized into four groups: newly diagnosed FNAIT cases (*n* = 9, Group 1), antenatally IVIg-treated FNAIT cases (*n* = 14, Group 2), and controls with and without anti-HLA class I antibodies (*n* = 10, Group 3 and *n* = 10, Group 4, respectively). Only FNAIT cases caused by anti-HPA-1a antibodies were included. Newly diagnosed FNAIT cases (Group 1) were identified upon diagnostic testing at Sanquin Diagnostics, Amsterdam, The Netherlands. Cases were selected from a cohort of 77 cases diagnosed between January 2006 and January 2017; 22% of the newborns with newly diagnosed anti-HPA-1a-mediated FNAIT were small for gestational age [9]. Pathology departments were contacted to request placenta material. All cases with available material were included. In total, nine placentas of newly diagnosed FNAIT cases were available and could be included in Group 1. These placentas had initially been evaluated for other clinical reasons: small for gestational age (SGA) (*n* = 3), fetal distress during delivery (*n* = 2), a previous mola pregnancy (*n* = 1), suspected abruptio placentae (*n* = 1), premature delivery (*n* = 1), and a suspected congenital infection (*n* = 1). Group 2 consisted of 14 placentas from pregnancies complicated by FNAIT of which mothers received IVIg treatment at the Leiden University Medical Center (LUMC). Cases were identified because they were included in a randomized trial comparing low-dose IVIg (0.5 g/kg/week, *n* = 7, 10 samples) versus standard-dose IVIg (1 g/kg/week, *n* = 7, 11 samples) [27].

Twenty controls were selected from a cohort of cases of uncomplicated pregnancies that resulted in the delivery of a healthy child at the obstetric department of the LUMC or in the affiliated hospitals in the region. Placentas were stored after informed consent for research purposes; furthermore, storage of the placenta material, HLA typing, and HLA antibody screening took place for all these cases. Although the role of anti-HLA class I antibodies in FNAIT is the subject of scientific debate, anti-HLA class I reactive antibodies can bind to platelets, immune cells, and endothelial cells in the placenta [28]. Ten controls were selected based on the presence of anti-HLA class I type antibodies and a newborn positive for the targeted antigens; these cases were categorized as Group 3. Ten controls were selected based on the absence of anti-HLA class I or II antibodies and placed in Group 4. All mothers in Group 3 and 4 were genotyped and were HPA-1a-positive.

### 4.2. Clinical Data Collection and Definitions

Clinical data concerning obstetric and neonatal treatment were collected from the medical records. A low placental weight was defined as a weight below the 10th percentile [29]. Small for gestational age (SGA) was defined as a birthweight below the 10th percentile according to the Dutch reference curves for birthweight [30]. Clinical data collection was performed separately from laboratory experiments; data were de-identified and linked to a study number by an independent research nurse.

### 4.3. Ethics

Ethical approval was provided by the medical ethical committee of Leiden, Delft, The Hague, according to study protocol B18.033 for cases and protocol P13.084 for the controls. Placentas and serum samples of control cases were collected after informed consent.

### 4.4. HPA Antibody Detection

FNAIT was diagnosed by the presence of fetal or neonatal thrombocytopenia and/or bleeding symptoms in presence of fetal-maternal HPA incompatibility and HPA alloantibodies. HPA incompatibility was confirmed by maternal, paternal, and/or neonatal genotyping. HPA alloantibody screening was performed using PIFT (platelet immunofluorescence test) and MAIPA (monoclonal antibody immobilization of platelet antigens assay), as described by Porcelijn et al. [31].

### 4.5. HLA Antibody Detection and HLA Typing

Maternal anti-HLA antibodies were detected and typed using the Luminex Single bead Antigen assay (Lifecodes, Immucor, Norcross, GA, USA) for HLA class I and II at either the Department of Immunogenetics, Sanquin, Amsterdam (Group 1 and 2), or the Department of Immunology at the LUMC (Group 3 and 4). If HLA antibodies were present in the maternal serum, low-resolution genotyping was performed using PCR-SSP and sequence-based typing to determine whether or not the maternal HLA alloantibodies were child-specific.

### 4.6. Histopathology

Formalin-fixed paraffin-embedded (FFPE) tissues (one slide of the umbilical cord, one of the membranes, and at least two sections of normal placental parenchyma) were H&E stained according to standardized protocol [32]. Histopathology was reviewed according to the Amsterdam criteria by two experienced perinatal pathologists (P.G.J.N. and L.E.v.d.M.). The pathologists were blinded to clinical information, except for gestational age.

### 4.7. Immunohistochemistry

The presence of complement proteins was investigated using immunohistochemistry for C1q, C4d, membrane attack complex (MAC, SC5b-9), and mannose-binding lectin (MBL). Detailed protocols are reported in Appendix A. In brief, sections were deparaffinized and antigen retrieval was performed. After blocking for endogenous peroxidase, the sections were incubated with the primary antibody. Binding of the primary antibody was visualized with the appropriate secondary antibody (Dako Envision+, DakoCytomation, Glostrup, Denmark) and diaminobenzidine as a chromogen. Isotype-specific controls were used as negative controls (Appendix A). Sections were counterstained with hematoxylin.

### 4.8. Quantification of Immunohistochemical Staining

The immunohistochemical staining was scored semi-quantitatively as either absent (<10%), focal (10–50%), or diffuse (>50%) by two blinded observers (T.W.d.V. and R.D.M. v.B.). In case of discrepancy between the scores, consensus was reached in the presence of an independent blinded third researcher (M.B.). Examples for the semi-quantitative score can be found in Appendix A.

### 4.9. Statistical Analysis

Data analysis was performed using IBM SPSS Statistics 25.0 (Chicago, IL, USA). Descriptive statistics were used to report clinical characteristics by proportions and medians with interquartile ranges. Quantitative clinical characteristics were compared by the Kruskal–Wallis analysis. Categorical parameters were compared by the Chi-square analysis or Fisher Exact Test as appropriate. An ordinal logistic regression model was used for the comparison of complement deposits in the placenta. This statistical model was chosen because, in some FNAIT cases, only one sample was available instead of two.

## 5. Conclusions

Our findings support the hypothesis that HPA-1a antibodies affect placental function and may impair fetal growth. We demonstrated that increased classical pathway-mediated complement activation was present in placentas of pregnancies complicated by FNAIT. To what extent HPA-1a antibodies binding to trophoblast cells leads to placenta dysfunction should be further explored.

## Figures and Tables

**Figure 1 ijms-22-06763-f001:**
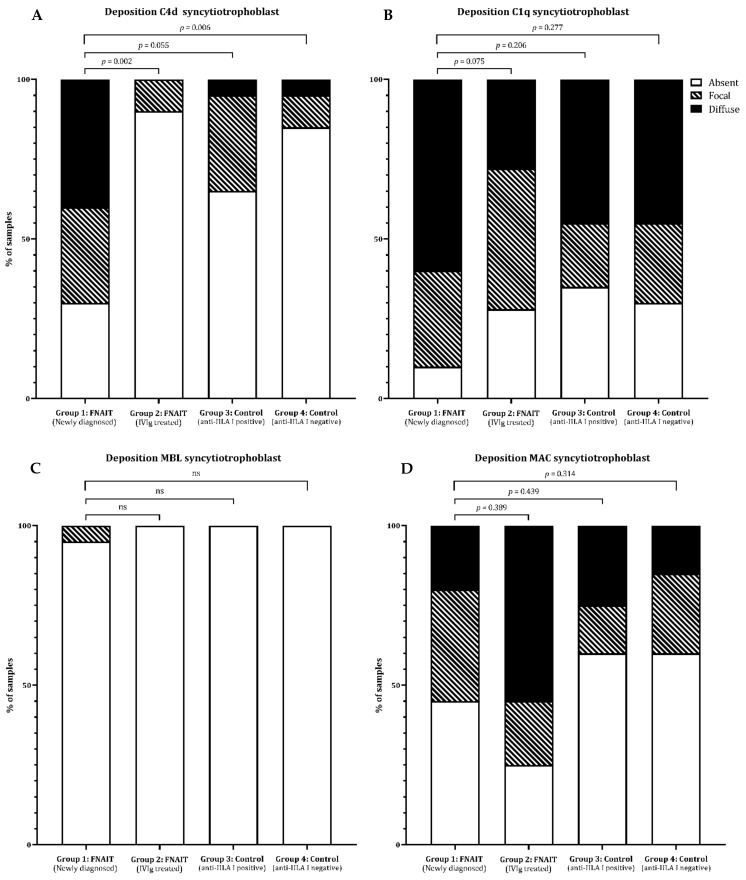
Semi-quantitative scoring of complement deposition syncytiotrophoblast. All complement depositions shown by immunohistochemistry at the syncytiotrophoblast were scored semi-quantitatively as absent (<10%), focal (10–50%), or diffuse (>50%). Figure (**A**) summarizes the scoring of C4d deposition at the syncytiotrophoblast. In Figure (**B**), scoring of C1q deposition, in Figure (**C**), scoring of MBL deposition, and in Figure (**D**), scoring of MAC deposition are summarized. Ordinal logistic regression was used to compare complement deposition score between the groups. FNAIT, fetal neonatal alloimmune thrombocytopenia; IVIg, intravenous immune globulins; HLA, human leukocyte antigen; MBL, mannose-binding lectin; MAC, membrane attack complex.

**Table 1 ijms-22-06763-t001:** Clinical characteristics.

	Group 1 FNAIT Cases Newly Diagnosed *n* = 9	Group 2 FNAIT Cases IVIg-Treated *n* = 14	Group 3 Controls Anti-HLA I Positive *n* = 10	Group 4 Controls Anti-HLA I Negative *n* = 10
Maternal characteristics				
Maternal age (years)—*median* (*IQR*)	31 (29–34)	33 (31–36)	33 (31–35)	33 (30–39)
Gravidity—*median* (*IQR*)	2 (1–2)	3 (2–4)	3 (2–3)	2 (1–4)
Parity—*median* (*IQR*)	0 (0–1)	1 (1–2)	1 (0–1)	1 (0–1)
Multiparous women—*n* (%)	6 (67%)	14 (100%)	9 (90%)	7 (70%)
Pre-eclampsia—*n* (%)	0	0	0	0
Delivery				
Spontaneous vaginal delivery—*n* (%)	1 (11%)	8 (57%)	4 (40%)	4 (40%)
CS fetal distress—*n* (%)	3 (33%)	1 (7%)	0	0
GA at delivery (weeks^+days^) **—*median* (*IQR*)	37^+1^ (33^+5^–40^+5^)	38^+3^ (37^+2^–38^+6^)	39^+4^ (39^+1^–40^+4^)	37^+1^ (38^+4^–40^+2^)
Neonatal data				
Sex (male)—*n (%)*	5 (56%)	7 (50%)	5 (50%)	5 (50%)
Birthweight (grams) **—*median (IQR)*	2405 (2099–3535)	3133 (2674–3493)	3700 (3319–3805)	3323 (3214–3814)
Small for gestational age *—*n (%)*	4 (44%)	1 (7%)	0	0
Skin bleeding only *—*n (%)*	4 (44%)	0	0	0
Intracranial hemorrhage *—*n (%)*	2 (22%)	1 (7%) ^Ф^	0	
Perinatal asphyxia—*n (%)*	1 (11%)	0	0	0
Platelet count nadir (×10^9^/l) **—*median* (*IQR*)	17 (9–43)	64 (21–170)	NT	0
HPA alloantibodies Mother HPA-1a-negative—*n* (%) Fetus HPA-1a-positive—*n* (%) Antibodies directed against; HPA-1a—*n* (%) HPA-1a and 3a—*n* (%) HPA-1a and 5b—*n* (%)	9 (100%) 9 (100%) 6 (67%) 1 (11%) 2 (22%)	14 (100%) 14 (100%) 14 (100%)	0 NT NT	0 NT NT
Anti-HLA class I present fetus-specific ^†^—*n* (%)	4 (57%), 2 missing	4 (44%), 5 missing	10 (100%)	0
Placenta characteristics ¶	*n =* 9	*n =* 9	*n =* 7	*n =* 8
Placenta weight (grams)—*median* (*IQR*)	460 (268–636)	500 (440–577)	572 (420–790)	610 (533–730)
Placenta weight < p10 *—*n* (%)	4 (44%)	1 (7%)	0	0

FNAIT, fetal neonatal alloimmune thrombocytopenia; IVIg, intravenous immune globulins; HLA, human leukocyte antigen; CS, caesarean section; GA, gestational age; l, liter; HPA, human platelet antigen p, percentile. * Data show a statistically significant difference (*p* < 0.05) when compared with the pooled controls by the Chi-Square test. ** Data show a statically significant difference (*p* < 0.05) when compared with the controls by Kruskal–Wallis analysis. ^Ф^ Intracranial hemorrhage developed at 27 weeks’ gestational age before administration of intravenous immune globulins to the mother was started. ^†^ Assessed in 7/9/10/10 cases, missing values for 7 (16%) cases due to lack of serum/DNA. ¶ Assessed in 9/9/7/8 cases, missing values for 10 (23%) cases.

**Table 2 ijms-22-06763-t002:** Placenta histopathology.

Pathology	Group 1 FNAIT Cases Newly Diagnosed *n* = 9	Group 2 FNAIT Cases IVIg-Treated *n* = 14	Group 3 Controls Anti-HLA I-Positive *n* = 10	Group 4 Controls Anti-HLA I-Negative *n* = 10
**Maturation**				
Delayed	4 (44%)	5 (36%)	1 (10%)	0
Corresponding with GA	3 (33%)	9 (64%)	9 (90%)	10 (100%)
Accelerated	2 (22%)	0	0	0
**Maternal vascular malperfusion**	**3 (33%)**	**3 (21%)**	**0**	**0**
Retroplacental hematoma	1	0	0	0
Infarction	3	0	0	0
Ischemia	0	2	0	0
Distal villous hypoplasia	1	3	0	0
**Fetal vascular malperfusion**	**1 (11%)**	**0**	**0**	**3 (30%)**
Avascular villi	1	0	0	3
**Ascending intrauterine infection**	**2 (22%)**	**1 (7%)**	**1 (10%)**	**2 (20%)**
Stage 1	2	1	0	2
Fetal response	0	0	1	0
**Villitis of unknown etiology**	**3 (33%)**	**3 (21%)**	**2 (20%)**	**1 (10%)**
Low-grade focal	3	3	1	1
Low-grade multifocal	0	0	1	0
**Massive perivillous fibrin depositions**	**0**	**1 (7%)**	**0**	**0**
**Signs of fetal hypoxia**	**2 (22%)**	**1 (7%)**	**0**	**0**
Mild hypoxia	2	1	0	0
**Chorangiosis**	**0**	**0**	**2 (20%)**	**0**
**Meconium**	**1 (11%)**	**0**	**0**	**0**

FNAIT, Fetal neonatal alloimmune thrombocytopenia; IVIg, intravenous immune globulin; HLA, human leukocyte antigen; GA, gestational age.

## Data Availability

Data available on request due to privacy/ethical restrictions. Outcomes of immunohistochemistry and histopathology assessment per case can be found in Appendix A.

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
