# Peer review of "Placental Complement Activation in Fetal and Neonatal Alloimmune Thrombocytopenia: An Observational Study"

_ijms, 2021, doi:10.3390/ijms22136763_

Round 1

Reviewer 1 Report

The focus how alloimmune thrombocytopenia may be related to placental dysfunction and lower birth weight is relevant and scientifically interesting and with many unsolved questions. The idea of studying complement activation in relation to HPA-1a alloimmunization is therefore good. The paper is well written and the data is mostly well presented. There are, however, some significant concerns, mainly related to the design and use of control groups.

Design:

During an 11 year period from a Centre in The Netherlands which yearly handles a high volume of platelet alloimmunized pregnancies, only 9 placentas/ pregnancies were included in the case group of non-treated FNAIT pregnancies. This is a very low number. Also, as the authors also state, the placentas/ pregnancies were selected for histopathological evaluation for other clinical indications than FNAIT. Therefore, the selection bias is likely huge and such low number of cases strongly questions the representability and generalizability of their data. Even though the authors acknowledge the possibility of such bias, this does not remove these concerns. Further, the IVIg-treated group were selected prospectively and thus any difference between newly cases and IVIg-treated cases may be due to less severe FNAIT or treatment, which is not possible to dechipher. Finally, the control groups are suboptimal. The authors state that these were selected from healthy uncomplicated pregnancies. None of the controls were SGA or from fetal distressed deliveries. These controls are therefore not appropriate as relevant controls for this study. Optimal would be placentas from HPA-1bb women without anti-HPA-1a antibodies, or at least matched controls from similar rates of SGA pregnancies or children delivered by emergency caesarean section. This last option should be possible to obtain without performing a large prospective study.

Control group, methods:

The authors do not state how the pregnancies were identified as HLA class I antibody positive? When and why were these pregnancies tested for anti-HLA class I antibodies? The idea of sorting these pregnancies based on HLA antibodies is good, but since there were no major differences in results, maybe the 2 control groups could be added together and just briefly mention in text that HLA antibody status did not have different effects?

Histopathology:

The authors found no significant differences in various placental pathologies between FNAIT cases or controls. They highlight that 44% of the untreated FNAIT cases displayed delayed placental maturation, but in fact 33% of placentas had normal maturation and 22% had accelerated maturation. Also, the control group is not appropriate to assess whether HPA-1a alloimmunization is a contributor to delayed maturation, as the control group is not matched for other clinical factors known to be associated with delayed maturation, such as SGA. Considering the low number of cases, combined with the recruitment challenges described above, these results are therefore not convincing. The idea is novel and interesting, though, and if it could be shown that the degree of delayed maturation is increased in FNAIT pregnancies compared to comparable non-FNAIT pregnancies the data would be very interesting. It would also be relevant to include what is the frequency of findings of delayed maturation overall in placentas examined for any clinical indication and/ or SGA – is there any difference?

The possible links between how HPA-1a alloimmunization is connected to complement activation, which in turn may lead to delayed maturation, is discussed briefly, but since this is the main finding of the paper, it deserves to be discussed in more depth. And were the newborns from placentas with delayed maturation also SGA?

A previous study from France (Dubruc et al, Placenta 2016) which found a high frequency of various chronic placental inflammations in FNAIT pregnancies, especially VUE and chronic intervillositis, contrasts the findings in this study, where no cases of CIV were seen. The authors do not convincingly explain these contrasting results.

Anti-HPA-1a antibody binding: The binding of anti-HPA-1a antibody to syncytiotrophoblasts is nicely shown, however in order to demonstrate that the binding is HPA-1a specific a better negative control is needed, i.e. HPA-1bb placenta. Also, a positive control demonstrating general B3 integrin binding would be nice. The negative control is polyclonal IgG, whereas a monoclonal anti-HPA-1a antibody is used to demonstrate binding. It would be nice to see how polyclonal anti-HPA-1a antibody binds also. Solid demonstration of specific HPA-1a binding to syncytiotrophoblasts would increase the value of this work. Others, for instance Kumpel et al, have previously shown binding of anti-HPA-1a antibodies to the brush border of placenta. The novelty of the current findings should be discussed based on previous reports.

Anti-HPA-1a antibodies also bind endothelial cells. The effects on fetal growth may be linked through effects on endothelial cells and not through trophoblasts, this is yet unsolved. The authors did not find any differences in complement activation in fetal vessels, but do not include any data regarding binding of anti-HPA-1a antibodies to fetal endothelial cells. The authors may include in their discussion thoughts why fetal vessels do not seem to be complement activated.

Complement activation:

Novel and interesting. However, not so clear in the manuscript how the authors think this is related and relevant to the other findings. As the paper reads now, each finding is presented as a separate thing. The different pieces needs to be better integrated in order to make one story. The discussion of complement in the paper lacks this. MAC being the endpoint and final proof of full complement activation; how do the authors interpret that there were no differences between the 4 study groups in MAC deposition? I would like to see comparable control placentas; fx SGA placentas without FNAIT – if difference in complement activation is shown more in such FNAIT placentas the data would be much more convincing.

Most of the newly diagnosed FNAIT cases were delivered by CS, whereas most of the IVIg-treated pregnancies were vaginally delivered. The most convincing complement activation data differences were between these 2 groups. Could different routes of delivery affect these results?

33% VUE: How were the SGA pregnancies and complement activated placentas distributed; were they also VUE? These data is missing now.

In general; to improve the quality of the data at hand, I suggest to look at the SGA-pregnancies, complement activated pregnancies, delayed maturation pregnancies and VUE-pregnancies together. If these are the same pregnancies displaying these factors, the argument is much more convincing than presenting each factor isolated. What does it mean that complement activation is less in IVIg-treated pregnancies, but delayed maturation is similar – what are the implications and significance of these findings?? Also, I suggest to add more placentas to the control group including placentas from SGA and VUE-pregnancies without FNAIT, in order to have a more appropriate control group.

Author Response

REVIEWER 1

The focus how alloimmune thrombocytopenia may be related to placental dysfunction and lower birth weight is relevant and scientifically interesting and with many unsolved questions. The idea of studying complement activation in relation to HPA-1a alloimmunization is therefore good. The paper is well written and the data is mostly well presented. There are, however, some significant concerns, mainly related to the design and use of control groups.

Before we answer the questions of this reviewer we would like to thank this reviewer for carefully reading our paper. By providing valuable concerns and remarks we were able to improve our paper.

 (Comments of the reviewer were typed in bold, our answers were typed in italic. Adapted text from the manusctipt were typed in a smaller font size. We used line numbers from the manuscript with track changes.)

  1. Design:

During an 11 year period from a Centre in The Netherlands which yearly handles a high volume of platelet alloimmunized pregnancies, only 9 placentas/ pregnancies were included in the case group of non-treated FNAIT pregnancies. This is a very low number. Also, as the authors also state, the placentas/ pregnancies were selected for histopathological evaluation for other clinical indications than FNAIT. Therefore, the selection bias is likely huge and such low number of cases strongly questions the representability and generalizability of their data. Even though the authors acknowledge the possibility of such bias, this does not remove these concerns.

We would like to thank the reviewer for sharing his/her concerns about the study design. We agree with the reviewer that due to the study design selection could be present. From a cohort of 77 FNAIT cases with HPA-1a antibodies present in total 9 placentas could be obtained via pathology departments and included in this study. Clinical characteristics of these 77 cases were described in the supplemental data of this retrospective cohort study (Winkelhorst et al. Treatment and outcomes of fetal/neonatal alloimmune thrombocytopenia: a nationwide cohort study in newly detected cases, Br J Heamatol, 2018). Demographic and clinical characteristics of this retrospective cohort were comparable to our study group on gravidity, birthweight and platelet count nadir. Median gestational age was one week lower in our study group. There were 2 cases (22%) with ICH where a 8% rate was expected in immunised women and as the reviewer indicates this slight overrepresentation could be expected in this study design. (Link: https://onlinelibrary.wiley.com/action/downloadSupplement?doi=10.1111%2Fbjh.15216&file=bjh15216-sup-0001-Supinfo.docx) We agree with the reviewer that we have to be careful with our conclusions due to the possibility of bias. Information concerning the selection of this group was added to method section (line 265 – 267):

Only FNAIT cases caused by anti-HPA-1a antibodies were included. Newly diagnosed FNAIT cases (Group 1) were identified upon diagnostic testing at Sanquin Diagnostics, Amsterdam, The Netherlands. Cases were selected from a cohort of 77 cases diagnosed between January 2006 and January 2017; 22% of the newborns with newly diagnosed anti-HPA-1a mediated FNAIT were small for gestational age.

FNAIT is mainly diagnosed postnatally, thus placentas are often already disposed when FNAIT is diagnosed. Therefore it is challenging to collect placenta material of newly diagnosed FNAIT cases. Placentas of FNAIT complicated pregnancies that we included in our study were available at the pathology department for various reasons as mentioned in the method section (line 268 – 273):

 Pathology departments were contacted to request placenta material. All cases with available material were included. In total, nine placentas of newly diagnosed FNAIT cases were available and could be included in Group 1. These placentas had initially been evaluated for other clinical reasons; small for gestational age (SGA) (n = 3), fetal distress during delivery (n = 2), a previous mola pregnancy (n = 1), suspected abruptio placentae (n = 1), premature delivery (n = 1) and a suspected congenital infection (n = 1).  

Further, the IVIg-treated group were selected prospectively and thus any difference between newly cases and IVIg-treated cases may be due to less severe FNAIT or treatment, which is not possible to dechipher.

 We agree with the reviewer collection of placentas from IVIg-treated cases and newly detected FNAIT cases is different which may have impact on the distribution of disease severity and confounding factors in both groups. We included this as a remark in the discussion section (line 179 – 181):

 This difference in the inclusion of our groups make it difficult to decipher if differences in complement deposits were due less severe FNAIT or the effect of IVIg treatment.

 Finally, the control groups are suboptimal. The authors state that these were selected from healthy uncomplicated pregnancies. None of the controls were SGA or from fetal distressed deliveries. These controls are therefore not appropriate as relevant controls for this study.

 Complement deposition was assessed in placentas of children born SGA (Buurma et al. Preeclampsia Is Characterized by Placental Complement Dysregulation. Hypertension. 2012). In 14 children born SGA (birthweight percentile < 2.3) focal C4d deposition was present at the syncytiotrophoblast and absent in the other 11 cases. Presence of C4d was shown according to the same methods as used for this study. We agree with the reviewer that this subject needs more attention in our discussion section, therefore we emphasized this study our discussion section (line 187 - 189):

 Furthermore, Buurma et al. [16], found a comparable amount of C4d depositions in IUGR and control placentas. Thus, SGA was suggested not to be associated with C4d deposition in the placenta.

 We think that the study presented in this manuscript is an important stepping stone and indicates that this is an important to assess further research projects (line 253 – 256):

 Based on the observation of increased classical route complement activation in FNAIT placentas and histopathological findings it is certainly of interest to perform additional research in placentas of HPA-1a negative women with and without alloantibodies and infants born SGA.

 Optimal would be placentas from HPA-1bb women without anti-HPA-1a antibodies, or at least matched controls from similar rates of SGA pregnancies or children delivered by emergency caesarean section. This last option should be possible to obtain without performing a large prospective study.

 Inclusion of HPA-1bb women without antibodies as a control group would be interesting, however since only 2.4% of the Dutch population is HPA-1a negative, selection of this control group is challenging in the Netherlands. Unfortunately we do not have an HPA-1a negative placenta available. Currently, we are performing a prospective screening in the Netherlands. If this study is finished, it would perhaps be possible to select a cohort of HPA-1a negative women with and without HPA-1a antibodies to assess the histopathology in their placentas. (Winkelhorst et al., HIP (HPA-screening in pregnancy) study: protocol of a nationwide, prospective and observational study to assess incidence and natural history of fetal/neonatal alloimmune thrombocytopenia and identifying pregnancies at risk. BMJ open, 2020). Based on the comments of the reviewer we have added this suggestion to the discussion section (line 253 – 256):

 Based on the observation of increased classical route complement activation in FNAIT placentas and histopathological findings it is certainly of interest to perform additional research in placentas of HPA-1a negative women with and without alloantibodies and infants born SGA.

  1. Control group, methods:

The authors do not state how the pregnancies were identified as HLA class I antibody positive? When and why were these pregnancies tested for anti-HLA class I antibodies? The idea of sorting these pregnancies based on HLA antibodies is good, but since there were no major differences in results, maybe the 2 control groups could be added together and just briefly mention in text that HLA antibody status did not have different effects?

 Anti-HLA class I antibodies are often present in FNAIT cases, and these antibodies can possibly bind to other cells like endothelial cells and immune cells in the placenta. We agree with the reviewer that results of the groups with and without HLA antibodies were similar. Although we agree with the reviewer that it might be possible to combine the control groups, we would like to keep the groups as they were presented in the manuscript because the role of anti-HLA class I in FNAIT is controversial. (Dahl et al. A combined effect of anti-HPA-1a and anti-HLA Class I in pregnancy? Transfusion 2020)

 The placentas in this group were selected from uncomplicated pregnancies. Materials and clinical data were stored anonymously for research purposes after informed consent after ethical approval. For the use in other studies, the presence of anti-HLA antibodies was assessed in the maternal serum in a large cohort. In addition fetal and maternal HLA typing took place to check if these antibodies were child-specific. We added a sentence to further clarify the selection of this cohort (line 278 – 288):

 Twenty controls were selected from a cohort of cases of uncomplicated pregnancies that resulted in the delivery of a healthy child at the obstetric department of the LUMC or in the affiliated hospitals in the region. Placentas these cases were stored after informed consent for research purposes, besides storage of the placenta material, HLA typing and HLA antibody screening took place for all these cases. Although the role of anti-HLA class I antibodies in FNAIT is subject of scientific debate, anti-HLA class I reactive antibodies can bind to platelets, immune cells and endothelial cells in the placenta. Ten controls were selected based on the presence of anti-HLA class I type antibodies and a newborn positive for the targeted antigens, these cases were allocated in Group 3. Ten controls were selected based on the absence of anti-HLA class I or II antibodies and allocated in Group 4. All mothers in Group 3 and 4 were genotyped and were HPA-1a positive.

  1. Histopathology:

The authors found no significant differences in various placental pathologies between FNAIT cases or controls. They highlight that 44% of the untreated FNAIT cases displayed delayed placental maturation, but in fact 33% of placentas had normal maturation and 22% had accelerated maturation. Also, the control group is not appropriate to assess whether HPA-1a alloimmunization is a contributor to delayed maturation, as the control group is not matched for other clinical factors known to be associated with delayed maturation, such as SGA. Considering the low number of cases, combined with the recruitment challenges described above, these results are therefore not convincing. The idea is novel and interesting, though, and if it could be shown that the degree of delayed maturation is increased in FNAIT pregnancies compared to comparable non-FNAIT pregnancies the data would be very interesting. It would also be relevant to include what is the frequency of findings of delayed maturation overall in placentas examined for any clinical indication and/ or SGA – is there any difference?

 We would like to thank the reviewer for his/her kind words and the acknowledgement of finding our work novel and interesting. We agree with the reviewer that the presence of placentas with a delayed maturation pattern the group with newly detected HPA-1a alloimmunisation cannot be seen than more than a suggestion of an association. We agree that it is important to relate the pathology findings to the clinical characteristics in our study. Delayed maturation can be seen in placentas affected by villitis or in association with gestational diabetes. Delayed maturation can lead to fetal hypoxia and distress. (Jaiman et al., Placental delayed villous maturation is associated with evidence of chronic fetal hypoxia. Journal of perinatal medicine 2020) In the 4 newly diagnosed FNAIT cases born SGA delayed maturation was present in only one case, accelerated in one case and corresponding to gestational age in the other two. We added this to the result section (line 145-148):

In three of the nine cases (Group 1 and 2; 3/9 cases; 33%) with a delayed maturation, the presence of C4d at the syncytiotrophoblast was detected. In one placenta of the newly di-agnosed FNAIT cases with delayed maturation the infant was born SGA.

We emphasized that these differences were not statistically significant in the discussion (line 168 – 170):

 Histopathological examination showed delayed maturation of the placenta in 44% of the newly diagnosed FNAIT cases, 36% of the IVIg-treated FNAIT cases and only in one of the controls (10%) (NS).

 Furthermore we discussed the clinical relevance of these findings and improved in the discussion section (line 236 – 240):

 Delayed placental maturation was found in 44% of the newly diagnosed, 36% of the IVIg treated FNAIT cases and only once in the control group (10%). Delayed maturation might be caused by villitis by the binding of alloantibodies to trophoblasts altering growth and maturation of the terminal villi. Clinical symptoms related to delayed maturation are fetal hypoxia. Delayed maturation was also found in the placentas of β3-immunised mice.

The possible links between how HPA-1a alloimmunization is connected to complement activation, which in turn may lead to delayed maturation, is discussed briefly, but since this is the main finding of the paper, it deserves to be discussed in more depth. And were the newborns from placentas with delayed maturation also SGA?

To answer briefly to the question of the reviewer; we do not find an association between the presence of delayed maturation and SGA in our cohort. We added this as a remark to our manuscript in (147 – 148):

 In one placenta of the newly diagnosed FNAIT cases with delayed maturation the infant was born SGA.

A previous study from France (Dubruc et al, Placenta 2016) which found a high frequency of various chronic placental inflammations in FNAIT pregnancies, especially VUE and chronic intervillositis, contrasts the findings in this study, where no cases of CIV were seen. The authors do not convincingly explain these contrasting results.

We agree with the reviewer that the contrasting findings were mentioned but not explained convincingly. Gestational age between these groups were different. We added this reasoning to our discussion section (line 243-247):

 In other cohort studies that assess the histopathology of the placenta in FNAIT, chronic villitis, chronic chorioamnionitis and chronic intervillositis were observed. Aberrant histopathology was less pronounced in our FNAIT cases compared to the findings in these studies. However, median gestational age at delivery was 34 weeks in the French study and 37 weeks in our study which may explain the differences in histopathology.

Anti-HPA-1a antibody binding: The binding of anti-HPA-1a antibody to syncytiotrophoblasts is nicely shown, however in order to demonstrate that the binding is HPA-1a specific a better negative control is needed, i.e. HPA-1bb placenta. Also, a positive control demonstrating general B3 integrin binding would be nice. The negative control is polyclonal IgG, whereas a monoclonal anti-HPA-1a antibody is used to demonstrate binding. It would be nice to see how polyclonal anti-HPA-1a antibody binds also. Solid demonstration of specific HPA-1a binding to syncytiotrophoblasts would increase the value of this work. Others, for instance Kumpel et al, have previously shown binding of anti-HPA-1a antibodies to the brush border of placenta. The novelty of the current findings should be discussed based on previous reports.

We agree with the reviewer that an HPA-1a negative placenta would have been the optimal control, unfortunately an HPA-1a negative placenta is not present in our collection. We have added this as a remark to the discussion section (line 204 - 206):

 The immunohistochemical staining that showed binding of anti-HPA-1a to the placenta could also be evaluated in an HPA-1a negative placenta to further evaluate the specificity of or staining. In the current experiment we used polyclonal IgG as a negative control.

 In line with the comments of the reviewer we adapted the paragraph of our discussion section (line 194 – 203)  that describe our findings in comparison of those of others. (Yougbaré et al. Activated NK cells cause placental dysfunction and miscarriages in fetal alloimmune thrombocytopenia. Nature communications 2017;  Kumpel et al., Ultrastructural localization of glycoprotein IIIa (GPIIIa, beta 3 integrin) on placental syncytiotrophoblast microvilli: implications for platelet alloimmunization during pregnancy. Transfusion 2008; Eksteen et al., Anti-human platelet antigen (HPA)-1a antibodies may affect trophoblast functions crucial for placental development: a laboratory study using an in vitro model. Reproductive biology and endocrinology 2017)

 We visualised binding of anti-HPA-1a to the syncytiotrophoblast. Binding of anti-HPA-1a is in line with studies that showed the expression of the β3-integrin on syncytiotrophoblast and extravillous trophoblasts. In these studies an antibody directed against the β3 integrin was used whereas in our study we used a monoclonal antibody directed against the β3 integrin antigen HPA-1a.Eksteen et al. also showed binding of the anti-HPA-1a antibody to the αVβ3 receptor on isolated trophoblasts. They confirmed previous findings with endothelial cells that this binding leads to impaired functioning of the receptor, with a reduced ability of the cultured cells to migrate and adhere.

Anti-HPA-1a antibodies also bind endothelial cells. The effects on fetal growth may be linked through effects on endothelial cells and not through trophoblasts, this is yet unsolved. The authors did not find any differences in complement activation in fetal vessels, but do not include any data regarding binding of anti-HPA-1a antibodies to fetal endothelial cells. The authors may include in their discussion thoughts why fetal vessels do not seem to be complement activated.

 We agree with the reviewer that our thoughts about this subject should be added to the discussion. We can only speculate why we did not find differences in complement activation. We added this paragraph to our discussion (line 227 - 234):

 It is known that subtypes of HPA-1a directed antibodies can bind to endothelial cells. In contrast to the differences in complement depositions between the groups at the syncytiotrophoblast no differences were observed in complement depositions between the fetal vessels of the FNAIT cases and controls. Several factors can explain these different ob-servations the syncytiotrophoblast and fetal vessels; lower fetal antibody titres in the fetus, lower availability of complement proteins at the side of fetal vessels or differences in com-plement regulatory proteins at the surface of endothelial cells and syncytiotrophoblast.

  1. Complement activation:

Novel and interesting. However, not so clear in the manuscript how the authors think this is related and relevant to the other findings. As the paper reads now, each finding is presented as a separate thing. The different pieces needs to be better integrated in order to make one story. The discussion of complement in the paper lacks this. MAC being the endpoint and final proof of full complement activation; how do the authors interpret that there were no differences between the 4 study groups in MAC deposition? I would like to see comparable control placentas; fx SGA placentas without FNAIT – if difference in complement activation is shown more in such FNAIT placentas the data would be much more convincing.

We agree with the reviewer that the integration of separate findings needed to be improved. We adapted the discussion section and in particular the last paragraph of the subheading 2. Interpretation. In this paragraph the integration of complement activation, histopathology and clinical characteristics were discussed.

 We think that C4d can be seen as a footprint of classic route complement activation whereas MAC is only temporarily present as mentioned in the discussion section (line 219 – 226):

 We did not observe increased staining for MAC at the syncytiotrophoblast in the newly diagnosed FNAIT cases. This may be explained by the notion that the complement system is tightly regulated in the placenta by complement inhibitory proteins. For instance, membrane cofactor protein (CD46), decay-accelerating protein (CD55) and MAC-inhibitory protein (CD59) inhibit continuation of the complement cascade beyond C4b formation. C4d is seen as a footprint of complement activation, whereas MAC deposition can be transient. Moreover, C4d is an accepted biomarker in antibody-mediated transplant rejection and also acknowledged in antibody-mediated pregnancy complications, to which HPA-1a now may be added.

As mentioned in our previous remarks C4d was not present in children with a low birthweight. (Buurma et al. Preeclampsia Is Characterized by Placental Complement Dysregulation. Hypertension. 2012). We improved this part of the discussion section (line 187 – 189) as mentioned before (third question chapter 1. Design).

Most of the newly diagnosed FNAIT cases were delivered by CS, whereas most of the IVIg-treated pregnancies were vaginally delivered. The most convincing complement activation data differences were between these 2 groups. Could different routes of delivery affect these results?

We think it is unlikely that the differences in complement activation between the newly diagnosed cases and IVIg treated cases could be explained by the delivery mode. In the newly diagnosed FNAIT group one case and in the IVIg treated group eight cases were delivered spontaneous vaginally. In the newly diagnosed FNAIT cases C4d deposition was present in cases delivered by CS whereas in the IVIg treated cases only in the two cases that were vaginally delivered. In addition no relationship between C4d deposition and delivery mode was observed in a similar study performed by Buurma et al. (Buurma et al. Preeclampsia Is Characterized by Placental Complement Dysregulation. Hypertension. 2012). We thank the reviewer for this suggestion and added this comment to our results section (line 121 - 122)

 No relationship was observed between the mode of delivery and C4d deposition at the syncytiotrophoblast.

 In addition we added this point to the discussion section (Line 193 – 194):

Comparable to our study, Buurma et al., also did not find an association between complement activation and mode of delivery was found.

33% VUE: How were the SGA pregnancies and complement activated placentas distributed; were they also VUE? These data is missing now.

Thank you for this remark. As mentioned previously we adapted the result section based on this comment (line 149 - 152):

 Low grade villitis of unknown etiology was observed in all groups. In all three cases with low grade villitis of unknown etiology in Group 1, C4d deposition at the syncytiotrophoblast was observed, these infants were all born SGA.

And to the discussion section (line 169 – 171):

 Low grade villitis of unknown etiology was present in all groups. In the placenta of three out of four newly diagnosed FNAIT born SGA both low grade villitis of unknown etiology and C4d deposition at the syncytiotrophoblast were observed.

 In addition we added this to our discussion section, also based on the next remark of the reviewer that the integration of data could improve the quality of this paper (line 241 - 245):

C4d deposition and low grade villitis of unknown etiology were both present in three out of four newly diagnosed FNAIT cases born SGA. This finding can point to a correlation between complement deposition and villitis with fetal growth restriction in FNAIT. However, our sample size was small and low grade villitis of unknown etiology was also present in some of the control cases, therefore thisese findings should be interpreted with caution.

In general; to improve the quality of the data at hand, I suggest to look at the SGA-pregnancies, complement activated pregnancies, delayed maturation pregnancies and VUE-pregnancies together.

If these are the same pregnancies displaying these factors, the argument is much more convincing than presenting each factor isolated. What does it mean that complement activation is less in IVIg-treated pregnancies, but delayed maturation is similar – what are the implications and significance of these findings?? Also, I suggest to add more placentas to the control group including placentas from SGA and VUE-pregnancies without FNAIT, in order to have a more appropriate control group.

We would like to thank the reviewer. Bases on the concerns of the reviewer about the selection of our study groups and the controls, we discussed the selection our cases and controls in this letter and adapted the discussion section.

 We agree with the reviewer that the integration of different findings could together could be improved. Therefore, we improved the discussion section and integrated the results of complement activation, histopathology with the clinical characteristic to improve the quality of the paper.

Reviewer 2 Report

The manuscript is overall well written and clearly presented. The topic is interesting and the investigations carried out are thoroughly explained.

There are only few minor revisions:  

Abstract: Binding of anti-HPA-1a to the syncytiotrophoblast was demonstrated 
: where? All groups? 

Introduction: - the presence of anti-HPA-1a : where? Placenta? Which cells?

Author Response

REVIEWER 2

The manuscript is overall well written and clearly presented. The topic is interesting and the investigations carried out are thoroughly explained.

We would like to thank the reviewer for his/her kind words and suggestions to improve our manuscript.

 (Comments of the reviewer were typed in bold, our answers were typed in italic. Adapted text from the manusctipt were typed in a smaller font size. We used line numbers from the manuscript with track changes.)

There are only few minor revisions: 

Abstract: Binding of anti-HPA-1a to the syncytiotrophoblast was demonstrated: where? All groups?

Binding of anti-HPA-1a to the syncytiotrophoblast was demonstrated in an HPA-1a positive placenta of an uncomplicated pregnancy without HLA class I antibodies. We clarified this sentence (line 32):

Binding of anti-HPA-1a to the syncytiotrophoblast was demonstrated in a placenta from an uncomplicated pregnancy.

Introduction: - the presence of anti-HPA-1a : where? Placenta? Which cells?

Anti-HPA-1a was present in maternal serum. We clarified this sentence (line 63):

 In a Norwegian cohort study, the presence of anti-HPA-1a in maternal serum was associated with a reduced birthweight in infants diagnosed with FNAIT, which was also reported in other cohorts.

Round 2

Reviewer 1 Report

Dear authors, thank you for the your comments and revised manuscript. The discussion has improved. The wording and language of the new text added especially in the discussion has several mistakes, please read carefully and correct.

I have one remaining concern with this manuscript, which is regarding the demonstrating of anti-HPA-1a binding to syncytiotrophoblasts. The authors chose not to perform any of the additional experiments that I suggested in my first review. As the data stands now, the binding of anti-HPA-1a to placenta is not convicingly demonstrated. Primarily, since an optimal negative control using HPA-1bb placenta is missing, but also there is just 1 very zoomed-in picture of a placental villi where the brown color indicating antibody binding is also seen quite a lot inside the villi. And as mentioned previously, the negative control of polyclonal anti-s is not solid. 

Sound demonstration of anti-HPA-1a binding to syncytiotrophoblasts is important and would be novel data, therefore it should be presented with better positive and negative controls, and also preferably with section pictures with overview of more villi and not just one small villi. The story of complement activation, delayed maturation and also how the SGA cases/ VUE cases and complement activated placentas greatly overlap makes a nice story itself, and actually the antibody binding data does not add to this story. My suggestion would therefore be to remove these data from the manuscript, if the authors do not wish to pursue with more staining experiments at the moment, but maybe consider this for a future work? In that case, I also suggest to highlight to the abstract that the SGA cases/VUE/ complement cases greatly overlap, since this strengthens the hypothesis of a biological link and not just an association.
